# 12-month substance use disorders among first-year university students in Kenya

**Florence Jaguga**[1]*, **Muthoni Mathai**[2], **Caroline Ayuya**[3], **Ongecha Francisca**[4], **Catherine Mawia Musyoka**[2], **Jasmit Shah**[5], **Lukoye Atwoli**[5]

1 Department of Alcohol and Drug Abuse Rehabilitation Services, Moi Teaching and Referral Hospital, Eldoret, Kenya, 2 Department of Psychiatry, School of Medicine, College of Health Sciences, University of Nairobi, Nairobi, Kenya, 3 Department of Psychology & Counseling, Daystar University, Nairobi, Kenya, 4 Department of Medicine, Therapeutics, Dermatology & Psychiatry, School of Medicine, Kenyatta University, Nairobi, Kenya, 5 Brain and Mind Institute and the Department of Internal Medicine, Medical College East Africa, Aga Khan University, Nairobi, Kenya

* flokemboi@gmail.com

## Abstract

### Objectives

The period of entry into university represents one of vulnerability to substance use for university students. The goal of this study is to document the 12-month prevalence of substance use disorders among first year university students in Kenya, and to test whether there is an association between substance use disorders and mental disorders.

### Methods

This was a cross-sectional online survey conducted in 2019 and 2020 as part of the World Health Organization's World Mental Health International College Student (WMH-ICS) survey initiative. A total of 334 university students completed the survey. Descriptive statistics were used to summarize the demographic characteristics of the participants. Multivariate logistic regression was used to assess the association between substance use disorder and mental disorders after adjusting for age and gender.

### Results

The 12-month prevalence for alcohol use disorder was 3.3%, while the 12-month prevalence for other substance use disorder was 6.9%. Adjusting for age and gender, there was an association between any substance use disorder and major depression, generalized anxiety disorder, bipolar 1 disorder, intermittent explosive disorder, social anxiety disorder, suicidal ideation, suicide attempt, and non-suicidal self-injury.

### Conclusion

These findings highlight the need to institute policies and interventions in universities in Kenya that address substance use disorders and comorbid mental disorders among first-year students.

**Data Availability Statement:** All relevant data are within the paper and its Supporting Information files.

**Funding:** The authors received no specific funding for this work.

**Competing interests:** The authors have declared that no competing interests exist

**Abbreviations:** AOR, Adjusted Odds Ratio; CI, Confidence Interval; WMH-ICS, World Mental Health-International College Student survey.

## Introduction

The period of entry into university represents one of vulnerability to mental health and substance use problems for university students. Several reasons have been postulated for the high levels of mental health and substance use problems among this population including academic pressure, separation from family, transition to living independently, and managing life without parental supervision [1]. In addition, during this period, many students take on adult responsibilities e.g., working for the first time [1]. More importantly, late adolescence, corresponds to the age of onset for most severe mental illnesses [2] and is a peak period for initiation of substance use [3].

It is not surprising then that studies have shown that the first year of university is associated with high rates of harmful substance use. A study conducted in Northern Ireland among first year university students found that the prevalence of alcohol use disorder was 10.7% and that of other substance use disorder was 3.1%. In that study, the diagnosis of other substance use disorder was associated with being a male [4]. In a large cross-national survey (n = 14,371) conducted across eight countries (Australia, Germany, Spain, South Africa, Northern Ireland, United States, Mexico, Belgium), the 12-month prevalence of alcohol use disorder was 6.3%, and that of other substance use disorders was 3.0% among first year students [5]. In Kenya, Musyoka et al. [6] reported that the prevalence of daily alcohol use was 1%, and that of daily use of other substances ranged from 0–0.7% among first-year students at University of Nairobi.

College students who use substances often experience a myriad of problems. One main one is poor educational performance and attainment [7, 8]. Mojtabai et al. [7] analyzed a large dataset collected as part of the National Comorbidity Survey and found that mental illness including substance use among college students in the USA was associated with lower odds of college graduation. Tembo et al. [8] in a study conducted among Australian university students found that harmful alcohol use was linked to increased odds of psychological distress, being late for class, missing classes, inability to concentrate in class, and inability to complete assignments. A study conducted by Atwoli et al. [9] in Kenya found that students who used substances reported problems such as unprotected sex, sex that they regretted the next day, scuffles, loss and damage to property, and quarrels. Even worse is that studies show that substance use during college years persists beyond college [10]. Addressing substance use among college students is therefore of high priority. The World Mental Health-International College Student (WMH-ICS) survey initiative was launched in 2012 with the aim of systematically documenting the burden of mental and substance use disorders among first-year college students to institute appropriate preventive and treatment interventions. As part of this initiative, an online survey to investigate the burden of mental and substance use disorders among first-year university students was conducted across three Kenyan universities. The goal of this paper is to report on the prevalence of substance use disorders among this population. Prior papers emanating from studies conducted as part of the WMH-ICS survey, have focused on other mental disorders to the exclusion of substance use disorders [11]. Further, in this paper, the authors describe the association between substance use disorders and mental disorders among first year College students, an association that has not been reported in prior studies done in Kenya [6]. This paper aligns with target 3.5 of the Sustainable Development Goals which requires that treatment and prevention for substance use disorders is strengthened [12].

## Materials and methods

### Study setting

The data used for this study was derived from an online cross-sectional survey investigating the prevalence and correlates of mental disorders among first-year university students in

three universities in Kenya i.e., Moi University, University of Nairobi, and Daystar University.

Moi University is a public university in Kenya. It has a total enrollment of about sixty thousand students. It has 15 schools that offer both science and arts-based courses as well as campuses and constituent colleges spread across the country. The university admits about four thousand first-year students annually. Admission into the University takes place in March (for the College of Health Sciences) and August (for the other Colleges) [13].

The University of Nairobi was the first university to be established in Kenya. It is a public university with a total enrollment of about eighty thousand students. It has several schools, institutes, faculties, and centers offering a broad range of science, technology, humanities, social sciences, and arts-based courses as well as campuses and constituent colleges across the country. The university admits about ten thousand first-year students annually. Intake happens thrice a year: between September and October, between December and January, and between April and May [14].

Daystar University is a Christian private university with a total enrollment of about eighty-four thousand students. The University has two campuses located in Nairobi. The University has seven schools, three institutes and four directorates offering courses in science and arts-based courses. The university admits about nine hundred first-year students annually. Admission into the University takes place in January, May, and August [15].

## Study participants

Eligible participants for the online survey were first-year university students at the 3 institutions who were 18 years of age and above.

## Study procedures

The investigators sought and obtained email addresses from the three universities for all first-year students registering in the universities in the 2019/2020 academic year. An email was sent to all students inviting them to the survey. Qualtrics software was used to send the invites and disseminate the survey questionnaire.

The invitation included a brief description of the survey and a clear statement indicating that participation would not affect their studies either positively or negatively. This statement was followed by a brief description of the study objective and a request for participation. Students who selected "I agree to participate and confirm that I am 18 years of age and above" were then directed to the consent form and subsequently to the questionnaire for those who consented to participate in the survey.

The contacts for mental health services were provided in each of the participating institutions, so that those who receive a positive screen for a mental disorder could seek help. The survey was sent out to a total of 11,787 first-year students (4609 [39.1%] female and 7178 [60.9%] male) who registered in the 2019/2020 academic year. Of these, 1863 entered the survey resulting in a response rate of 15.8% (number of those entering the survey divided by number of emails sent). A total of 1529 students were removed either because they failed quality control checks, or had a significant amount of missing data, or did not meet inclusion criteria e.g., entered their age as less than 18 years despite confirming that they were aged 18 years and above prior to entering the survey. This resulted in a final sample of 334 respondents.

The surveys were sent out on various dates for the different universities. The survey was sent out to Moi University College of Health Sciences on 21st May 2019. Reminders were sent via email every three days for one month until 20th June 2019. For all other Colleges of Moi University, the survey was sent out on 12th of October 2019. Reminders were sent via email every three days for one month until 13th of November 2019 when the survey was closed. For

Daystar and University of Nairobi, the surveys were sent out on 6[th] May 2020. Reminders were sent via email every three days for one month until the 7th of June when the survey was closed. Web-based surveys are usually marred by low response rates [16]. The reminders were sent in order to enhance response rates.

## Measures and variables

For this study, the World Mental Health College Student Survey questionnaire was used [5]. It is a self-report, questionnaire that is fully structured and assesses multiple aspects of mental health. An initial section of the survey instrument assesses demographic data including age (as a continuous variable), gender (male, female, other), and student status (part time degree, full time degree, non-degree, other). Lifetime and 12-month mental disorders, based on the fifth edition of the diagnostic and statistical manual, are assessed for depression, generalized anxiety disorder, panic disorder, bipolar disorder, post-traumatic stress disorder, intermittent explosive disorder, suicidality, alcohol use disorder, and other substance use disorder (cannabis, cocaine, Lysergic Acid Diethylamide [LSD], opioids, speed, ecstasy, prescription medication). Questions on lifetime mental disorders explored whether the student had ever experienced any mental health symptoms in their lifetime, while 12-month mental disorders explored the occurrence of mental health symptoms over the 12-month period prior to the survey. No identifying information such as the student numbers or names were collected. The survey instrument was programmed into Qualtrics, a secure, web-based software platform designed to support data capture and dissemination for online surveys.

For this study, the dependent variable was any 12-month substance use disorder (alcohol or other substance use disorder), and the independent variables included demographic and 12-month mental health characteristics.

**Statistical analysis.** Descriptive statistics were used to summarize the demographic characteristics of the participants, where categorical data was presented as frequencies and percentages and continuous data as means and standard deviations (SD) or medians and interquartile ranges (IQR). We used post-stratification weight based on distribution of males and females in each university (Nairobi, Moi, Daystar). Fishers exact test for categorical data, and Kruskal Wallis test for continuous data, were used in the bivariate analysis, to assess for the association between 12-month substance use disorder and socio-demographic factors as well as the association between 12-month substance use disorder and 12-month mental disorders. Furthermore, multivariate logistic regression was used to assess the association between 12-month substance use disorder and 12-month mental disorders after adjusting for age and gender. For the multivariate logistic regression model, the dependent variable was any 12-month substance use disorder, and the independent variables were all different 12-month mental disorders run separately, with adjusting for age and gender. All assumptions were considered while running the models. In all analyses a p-value less than 0.05 was considered significant.

## Ethics approval and consent to participate

The study was reviewed and approved by Moi University/Moi Teaching and Referral Hospital Institutional Research and Ethics Committee (Approval number 0003265). Informed written online consent was obtained from each participant prior to participation in the study.

## Results

### Participant characteristics

A total of 334 students participated in the study. Most of the participants were from the University of Nairobi. The mean age of the participants was 19.5 (SD = 1.4) years. More than half

**Table 1. Participant characteristics.**

| Variable | | Mean (SD) | Freq (%) |
|---|---|---|---|
| Age (years) | | 19.5 (1.4) | |
| University | Daystar University | | 27 (8.1%) |
| | Moi University | | 86 (25.7%) |
| | University of Nairobi | | 221 (66.2%) |
| Gender | Male | | 181 (54.2%) |
| | Female | | 150 (44.9%) |
| | Transgender | | 3 (0.9%) |
| Current Student Status | Full-time degree | | 328 (98.5%) |
| | Part-time degree | | 2 (0.6%) |
| | Non-degree | | 1 (0.3%) |
| | Other | | 3 (0.9%) |

(54.2%) of the participants were male and most of them (98.5%) had enrolled to college on a full-time basis (Table 1).

## 12-month prevalence alcohol use disorder and other substance use disorder

Eleven participants (3.3%) met criteria for a 12-month alcohol use disorder, and twenty-three (6.9%) had a 12-month substance use disorder (other than alcohol). Twenty-six participants (7.8%) met criteria for any substance use disorder (Table 2).

## Association between any 12-month substance use disorder and demographic characteristics of the participants

There were no gender or age differences between students who had and those who did not have either a 12-month alcohol use disorder or 12-month other substance use disorder (Table 3).

## Association between any 12-month substance use disorder and 12-month mental disorders

At bivariate analysis, there was a significant association between any 12-month substance use disorder and several 12-month mental disorders including major depressive disorder ($p = 0.018$), generalized anxiety disorder ($p = 0.001$), bipolar I disorder ($<0.001$), intermittent explosive disorder ($<0.001$), post-traumatic stress-disorder ($<0.001$), suicidal Ideation ($<0.001$), suicide attempt ($<0.001$), and non-suicidal self-injury ($<0.001$) (Table 4).

In multivariate analysis, the mental disorders significantly associated with increased odds of having any substance use disorder were: major depressive disorder (AOR = 2.897; 95%

**Table 2. 12-month prevalence of alcohol use disorder and other substance use disorder.**

| Substance use disorder | | Freq | % |
|---|---|---|---|
| Alcohol Use Disorder | No | 323 | 96.7% |
| | Yes | 11 | 3.3% |
| Other substance use disorder | No | 311 | 93.1% |
| | Yes | 23 | 6.9% |
| Any 12-month substance use disorder | No | 308 | 92.2% |
| | Yes | 26 | 7.8% |

**Table 3. Association between any 12 -month substance use disorder and demographic characteristics of the participants.**

| | | Any substance use disorder | | | | p value |
|---|---|---|---|---|---|---|
| | | No (n = 308) | | Yes (n = 26) | | |
| Age (years) (median [IQR]) | | 19.0 [19.0, 20.0] | | 19.0 [18.0, 21.0] | | 0.794 |
| Gender | Male | 169 | 54.9% | 12 | 46.2% | 0.174 |
| | Female | 137 | 44.5% | 13 | 50.0% | |
| | Other | 2 | 0.6% | 1 | 3.8% | |

ci = 1.150, 7.298; p<0.024), generalised anxiety disorder (AOR = 7.882; 95% CI = 2.139, 29.041; p = 0.002), panic disorder (AOR = 4.949; 95% CI = 1.190, 20.585; p = 0.028), bipolar I disorder (AOR = 7.447; 95% CI = 2.623, 21.142; p<0.001), intermittent explosive disorder (AOR = 4.466; 95% CI = 1.893, 10.539; p = 0.001), post-traumatic stress disorder (AOR = 5.589; 95% CI = 2.030, 15.389; p = 0.001), social anxiety disorder (AOR = 2.692; 95% CI = 1.183, 6.124; p = 0.018), suicidal ideation (AOR = 5.424; 95% CI = 92.292, 12.832; p<0.001), suicide attempt (AOR = 13.136; 95% CI = 4.407, 39.157; p<0.001), non-suicidal self-injury AOR = 11.932; 95% CI = 4.687, 30.376; p<0.001). (Table 5).

## Discussion

This study found that the 12-month prevalence for alcohol use disorder among first year university students was 3.3%. Our findings are lower than those reported in other studies conducted among first year university students elsewhere. In Northern Ireland, McLafferty et al.

**Table 4. Association between any substance use disorder and mental disorders.**

| 12-month mental disorder | | Any substance use disorder | | | | p Value |
|---|---|---|---|---|---|---|
| | | No (n = 308) | | Yes (n = 26) | | |
| major depressive disorder | No | 269 | 87.3% | 18 | 69.2% | 0.018 |
| | Yes | 39 | 12.7% | 8 | 30.8% | |
| generalized anxiety disorder | No | 301 | 97.7% | 21 | 80.8% | 0.001 |
| | Yes | 7 | 2.3% | 5 | 19.2% | |
| panic disorder | No | 300 | 97.4% | 23 | 88.5% | 0.046 |
| | Yes | 8 | 2.6% | 3 | 11.5% | |
| bipolar I disorder | No | 294 | 95.5% | 19 | 73.1% | <0.001 |
| | Yes | 14 | 4.5% | 7 | 26.9% | |
| bipolar II disorder | No | 307 | 99.7% | 25 | 96.2% | 0.150 |
| | Yes | 1 | 0.3% | 1 | 3.8% | |
| intermittent explosive disorder | No | 221 | 71.8% | 9 | 34.6% | <0.001 |
| | Yes | 87 | 28.2% | 17 | 65.4% | |
| post-traumatic stress-disorder | No | 182 | 59.1% | 5 | 19.2% | <0.001 |
| | Yes | 126 | 40.9% | 21 | 80.8% | |
| social anxiety disorder | No | 207 | 67.2% | 11 | 42.3% | 0.017 |
| | Yes | 101 | 32.8% | 15 | 57.7% | |
| suicide ideation | No | 236 | 76.6% | 10 | 38.5% | <0.001 |
| | Yes | 72 | 23.4% | 16 | 61.5% | |
| suicide attempt | No | 298 | 96.8% | 18 | 69.2% | <0.001 |
| | Yes | 10 | 3.2% | 8 | 30.8% | |
| non-suicidal self-injury | No | 287 | 93.2% | 14 | 53.8% | <0.001 |
| | Yes | 21 | 6.8% | 12 | 46.2% | |

**Table 5. Multivariate analysis of association between any 12-month substance use disorder and any 12-month mental disorders.**

| 12-month mental disorder | | Adjusted odds ratio (95% CI)* | p value |
|---|---|---|---|
| major depressive disorder | Yes | 2.897 (1.150, 7.298) | 0.024 |
| | No | 1.0 | |
| generalized anxiety disorder | Yes | 7.882 (2.139, 29.041) | 0.002 |
| | No | 1.0 | |
| panic disorder | Yes | 4.949 (1.190, 20.585) | 0.028 |
| | No | 1.0 | |
| bipolar I disorder | Yes | 7.447 (2.623, 21.142) | <0.001 |
| | No | 1.0 | |
| bipolar II disorder | Yes | 12.486 (0.720, 216.654) | 0.083 |
| | No | 1.0 | |
| intermittent explosive disorder | Yes | 4.466 (1.893, 10.539) | 0.001 |
| | No | 1.0 | |
| post-traumatic stress-disorder | Yes | 5.589 (2.030, 15.389) | 0.001 |
| | No | 1.0 | |
| social anxiety disorder | Yes | 2.692 (1.183, 6.124) | 0.018 |
| | No | 1.0 | |
| suicide ideation | Yes | 5.424 (92.292, 12.832) | <0.001 |
| | No | 1.0 | |
| suicide attempt | Yes | 13.136 (4.407, 39.157) | <0.001 |
| | No | 1.0 | |
| non-suicidal self-injury | Yes | 11.932 (4.687, 30.376) | <0.001 |
| | No | 1.0 | |

*The model is adjusted for Age and Gender. Dependent Variable is Yes, any 12-month substance use disorder

[4] found that the 12-month prevalence of alcohol use disorder was 10.7%. In a large cross-national survey (Australia, Germany, Spain, South Africa, Northern Ireland, United States, Mexico, Belgium), the 12-month prevalence for alcohol use disorder was reported as 6.3% [5]. From the current study, the 12-month prevalence for other substance use disorders was 6.9%. Other studies have posted lower 12-month prevalences of other substance use disorders among first year university students including 3.1% in Northern Ireland [4], and 3.0% in a cross-national survey [5]. The relatively lower rates of alcohol use disorder and higher rates of other substance use disorders in the current study compared to other settings may be due to socio-cultural variations in the consumption and availability of substances among young people across countries [17, 18].

There were no gender differences in the prevalence of either alcohol use disorder or other substance use disorders in our study. Prior studies conducted among university students have reported higher odds of substance use among male students compared to females [4, 9, 19]. Findings from the current study suggest that the trends in substance use among young people are changing and that young women are engaging in substance use as much as their male counterparts.

Findings from the current study show that any 12-month substance use disorder was associated with mood disorders, anxiety disorders, and suicidality. The association between any 12-month substance use disorder and mood disorders, anxiety disorders and suicidality has been well documented in the general population [20] as well as among student populations [21, 22]. This association between mental disorders and substance use disorders is likely linked to multiple factors. People with mental disorders often self-medicate with substances to relieve

their mental health symptoms [23]. Another possible reason is that substances increase the risk of developing mental illness. For example, a systematic review and meta-analysis reported that alcohol use disorder was associated with an increased risk of subsequent depressive symptoms [24].

## Implications for policy and practice

The presence of substance use disorders in our study population highlights the need to put in place programs and interventions that address early identification, prevention, and treatment of substance use problems among first year students in universities in Kenya. In addition, it is important that universities develop institutional policies that support these interventions. Further, universities need to allocate resources for the prevention, treatment, and management of substance use disorders among first year university students. Programs that could be implemented include an out-patient mental health service manned by mental health professionals such as psychologists and psychiatrists; a program educating students on substance use, mental health, and life-skills; and a drop-in centre where the students can spend time engaging in pro-social activities. Health education and life skills training programs, as well as brief mental health interventions can be feasibly delivered using digital means [25].

The higher rates for other substance use disorders compared to alcohol use disorder reported in this study is concerning and suggests easier availability of other substances, most of which are illegal. The government ought to strengthen enforcement of policies and legislation that regulate the production, sale, and availability of illegal substances in Kenya.

The association between substance use disorders and other mental disorders found in this study, highlights the importance of implementing programs that integrate interventions for both disorders for university students in Kenya. For example, screening for substance use disorders ought to be accompanied by concurrent screening for these mental disorders. Secondly, interventions that are implemented for substance use disorders should include components that address co-occurring mental disorders. Thirdly, mental health providers offering treatment services to students should have competencies that enable them to treat both substance use disorders and mental disorders.

This study is not without limitations. First, the response rate was low despite several reminders sent via email. Secondly, there was a lot of missing data. One potential reason for the incomplete responses was the length of the online survey, which on average took 40 minutes to complete. Because of the low response rates, and missing data, generalizability of findings to the student population within the three universities is limited. In a future survey, the authors will make sure that important variables are programmed as mandatory so that missing values are minimized for all variables of interest. Further, in a future survey, the authors will add a feature that prevents the rest of the questionnaire from loading if the age is indicated as less than 18 within the questionnaire. That way, the survey does not proceed for underage participants.

A second limitation is that there may be differences in the characteristics between those who opted to participate in the survey and those who did not respond or complete the survey. This again limits the generalizability of the study findings.

Nonetheless, our study provides important findings that may inform practice and policy. Future work should focus on a bigger sample, and developing and testing interventions that address substance use disorders among first-year university students in Kenya.

## Supporting information

**S1 Checklist. STROBE statement—checklist of items that should be included in reports of observational studies.**
(DOCX)

**S1 Raw data.**
(XLSX)

## Acknowledgments

The authors acknowledge the World Mental Health -International College Survey team at Harvard University for supporting with quality checks and weighting for the data.

## Author Contributions

**Conceptualization:** Florence Jaguga, Muthoni Mathai, Caroline Ayuya, Ongecha Francisca, Catherine Mawia Musyoka, Lukoye Atwoli.

**Data curation:** Florence Jaguga.

**Formal analysis:** Jasmit Shah.

**Investigation:** Florence Jaguga, Lukoye Atwoli.

**Methodology:** Florence Jaguga, Muthoni Mathai, Caroline Ayuya, Ongecha Francisca, Lukoye Atwoli.

**Project administration:** Florence Jaguga, Muthoni Mathai, Caroline Ayuya, Ongecha Francisca, Catherine Mawia Musyoka, Lukoye Atwoli.

**Resources:** Florence Jaguga, Lukoye Atwoli.

**Supervision:** Florence Jaguga, Lukoye Atwoli.

**Writing – original draft:** Florence Jaguga.

**Writing – review & editing:** Florence Jaguga, Muthoni Mathai, Caroline Ayuya, Ongecha Francisca, Catherine Mawia Musyoka, Jasmit Shah, Lukoye Atwoli.

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
