## [Decision Letter · Decision Letter 0]

18 Jun 2023

PONE-D-23-1217212-month substance use disorders among first-year university students in KenyaPLOS ONE

Dear Dr. Jaguga,

Thank you for submitting your manuscript to PLOS ONE. After careful consideration, we feel that it has merit but does not fully meet PLOS ONE’s publication criteria as it currently stands. Therefore, we invite you to submit a revised version of the manuscript that addresses the points raised during the review process.

We look forward to receiving your revised manuscript.

Kind regards,

Md. Saiful Islam, BPH, MPH

Academic Editor

PLOS ONE

Reviewers' comments:

Reviewer's Responses to Questions

**Comments to the Author**

1. Is the manuscript technically sound, and do the data support the conclusions?

Reviewer #1: Partly

Reviewer #2: Partly

2. Has the statistical analysis been performed appropriately and rigorously? 

Reviewer #1: I Don't Know

Reviewer #2: No

3. Have the authors made all data underlying the findings in their manuscript fully available?

Reviewer #1: Yes

Reviewer #2: Yes

4. Is the manuscript presented in an intelligible fashion and written in standard English?

Reviewer #1: Yes

Reviewer #2: Yes

5. Review Comments to the Author

Reviewer #1: Data availability

Is the data fully available? To my knowledge there are restrictions with regards to sharing data from the World Mental Health International College Student Initiative publicly. Maybe check with Harvard about this.

Abstract

I think it would be sufficient to say that significant associations were found between SUD and depression, generalized anxiety disorder etc. – rather than including p values.

Introduction

Line 67 – should say Northern Ireland rather than Ireland as that is where the study was conducted.

Line 93 – It is not good practice to say ‘we’.

As part of this initiative, we conducted an online survey was conducted to investigate the burden of mental and SUDs among first-year university students across three Kenyan universities.

I think that a lot of this research has already been done elsewhere so it is important here to point out what is novel about the study. Is this the first of its kind in Kenya? If so, highlight that. What might be different about the students in Kenya?

Methods

What do you mean by a parent online cross-sectional survey?

I don’t think that you can say that you had a response rate of 15.8% when so many had to be removed from the survey and then go on to say you had a completion rate of 17.9%.

1529 students were removed either because they failed quality control checks, or had missing data, or did not meet inclusion criteria e.g., entered their age as less than 18 years

This is a major concern of mine. Why were so many removed? Surely, those who were not eligible to take part should have been detected earlier, perhaps when they were consenting, before taking the survey? From what I know of the survey, the age of participant is asked first, and they are not able to take the survey if they say that they are under 18. If this was the case this needs to be explained clearer in this section. Also, could missing data not have been dealt with?

Lines 146 -157. I think this could be cut back somewhat.

Line 159 doesn’t read well. The survey used was the World Mental Health……

I don’t think it is necessary to go into details about the parts of the survey that you haven’t included in your analysis. It would be good however to be more specific about how alcohol use and other substance use disorders were measured (scales used)

Line 172 – no ‘we’ No identifying information was collected…..

Line 176 – For this study…..

Results

It is important to refer to the corresponding table when reporting the results.

It is good to see such a high rate of male participants as many studies tend to have more female participants than males. What is the gender breakdown of students in the colleges? Perhaps there are more males than females in general? I think it would be good to include this when describing the characteristics of the sample.

Also were weights not applied in the analysis to account for age and gender characteristics of the student population in the three universities? I know that the World Mental Health College Student Initiatives usually include weights in their studies? In the acknowledgements it mentions weights. If weights were used it should be mentioned in the methodology section.

Explain what exactly this is - 12-month substance use disorder (other than alcohol). Is it drug use? If so, describe in details what it covers. This could be described in the methods section.

It is important to also report the number/prevalence of those who had any SUD in the text. Perhaps you could include co-morbid prevalence in Table 2.

The results are somewhat confusing and I am not sure why you run some of the analysis that you did.

I think that it would be better to report the prevalence rates of mental health disorders for all participants and then compare these with the rates for those with SUD. This would be much more informative than what is presented in Table 4.

That is the rationale for both the bivariate model and the regression model and why is it described as a multivariate logistic regression? A multivariate logistic regression is a model with more than one dependent variable. It the methodology it states that the dependent variable is just Any SUD. If this is the case, it would be an ordinary binary logistic regression. Also were all the assumptions of a regression considered? This would be important given the small sample size.

It is really now clear how this model was run and further detail are required. Were all the IVs entered together? It is important to consider how this would impact on the findings.

When reporting findings from a logistic regression it is normal to comment on the odds ratios rather than the p value. For example, students with depression were nearly three time more likely to have SUD (OR= 2.897). It is also normal to use 1.0 rather than reference and then place this below the ORs and Cis.

Discussion

Again, it isn’t good practice to refer to ‘our’ study. Say this study or something similar. The first few lines are very repetitive of what was said in the introduction. It would be good to re-word.

273 – People rather than persons

275 – typo – remove of before mental illness

It would be good to include further implications of the study in your conclusion.

Reviewer #2: It is very important for diagnosis by psychiatrists for both mental disorders and substance-related disorders.

Thus, all data should be re-categorized and re-analyze to be problematic substance use and "depression", "anxiety", whatever the questionnaire should be interpreted.

6. PLOS authors have the option to publish the peer review history of their article (what does this mean?). If published, this will include your full peer review and any attached files.

Reviewer #1: No

Reviewer #2: **Yes: **Chonnakarn Jatchavala M.D.,MSc.

---

## [Author Response · Author response to Decision Letter 0]

11 Aug 2023

Response to reviewers’ and Editor’s comments

The authors thank the Editor and reviewers for the detailed review and comments.

A point-by-point response to the comments has been provided below.

Editor’s comments

Response: We have aligned our document with the formatting requirements of Plos one formatting requirements.

2. In your Data Availability statement, you have not specified where the minimal data set underlying the results described in your manuscript can be found. PLOS defines a study's minimal data set as the underlying data used to reach the conclusions drawn in the manuscript and any additional data required to replicate the reported study findings in their entirety. All PLOS journals require that the minimal data set be made fully available.

Response: We have attached the raw data as a Supporting File 1.

Reviewer #1: 

1. Data availability: Is the data fully available? To my knowledge there are restrictions with regards to sharing data from the World Mental Health International College Student Initiative publicly. Maybe check with Harvard about this.

Response: We have attached the raw data as a Supporting File 1.

NB: Line numbers indicated in the responses refer to those of the document with tracked changes.

Abstract

1. I think it would be sufficient to say that significant associations were found between SUD and depression, generalized anxiety disorder etc. – rather than including p values.

Response: Thank you for this comment. We have deleted the p values (line 46-49)

Introduction

2. Line 67 – should say Northern Ireland rather than Ireland as that is where the study was conducted.

Response: This has been corrected (line 65). 

3. Line 93 – It is not good practice to say ‘we’. As part of this initiative, we conducted an online survey was conducted to investigate the burden of mental and SUDs among first-year university students across three Kenyan universities.

Response: This has been corrected. (Line 91 and 93)

4. I think that a lot of this research has already been done elsewhere so it is important here to point out what is novel about the study. Is this the first of its kind in Kenya? If so, highlight that. What might be different about the students in Kenya?

Response: We have highlighted that our work is novel since work done as part of the WMH ICS has focused on other mental disorders to the exclusion of alcohol and drug use disorders. Further, our work explored the association between substance use disorders and mental disorders, and this has not been explored among College in Kenya (Line 94-99).

Methods

5. What do you mean by a parent online cross-sectional survey?

Response: The word parent has been removed (line 104). 

6. I don’t think that you can say that you had a response rate of 15.8% when so many had to be removed from the survey and then go on to say you had a completion rate of 17.9%.

Response: The completion rate to us meant the proportion of those completing the survey out of those who entered the survey. We have removed the completion rate to avoid any confusion to the reader. (line 149).

7. 1529 students were removed either because they failed quality control checks, or had missing data, or did not meet inclusion criteria e.g., entered their age as less than 18 years. This is a major concern of mine. Why were so many removed? Surely, those who were not eligible to take part should have been detected earlier, perhaps when they were consenting, before taking the survey? From what I know of the survey, the age of participant is asked first, and they are not able to take the survey if they say that they are under 18. If this was the case this needs to be explained clearer in this section. Also, could missing data not have been dealt with?

Response: The authors agree that 1529 is a large number to remove, and this is a study limitation. Concerning age, we had a question asking participants to confirm that they were 18 years and above before consenting. (see line 136-137)

However, some participants confirmed that they were above 18 years, consented, and entered the survey, but went ahead to indicate their age as less than 18 within the questionnaire and thus we had to exclude those. 

In a future survey, we should add a feature that prevents the rest of the questionnaire from loading if the age is indicated as less than 18 within the questionnaire. That way, the survey does not proceed for underage participants. 

Other reasons for removal of data included significant amounts of missing data e.g., many participants started the survey but abandoned it after filling out just demographics only. One of the main challenges in survey studies is incomplete responses, especially if variables of interest are not entered. Such incomplete responses thus have to be excluded. In a future survey, we will make sure that important variables are programmed as mandatory so that missing values are minimized for all variables of interest.

We have included missing data and recommendations provided above within the limitations section of the manuscript (Line 342-351).

8. Lines 146 -157. I think this could be cut back somewhat.

Response: The authors have removed some content to make it clearer. (line 150-160)

9. Line 159 doesn’t read well. The survey used was the World Mental Health……

Response: This sentence has been refined. (line 162-163)

10. I don’t think it is necessary to go into details about the parts of the survey that you haven’t included in your analysis. It would be good however to be more specific about how alcohol use and other substance use disorders were measured (scales used).

Response: The authors have removed information on parts of the survey that have not been included in this analysis. (line 167-169)

11. Line 172 – no ‘we’ No identifying information was collected…..

Response: This has been corrected. (line 180-181)

12. Line 176 – For this study…..

Response: This has been corrected. (line 184)

Results

13. It is important to refer to the corresponding table when reporting the results.

Response: We have corrected this. (lines 213, 220, 228, 247, 263)

14. It is good to see such a high rate of male participants as many studies tend to have more female participants than males. What is the gender breakdown of students in the colleges? Perhaps there are more males than females in general? I think it would be good to include this when describing the characteristics of the sample.

Response: Yes. The sample was mostly male. We have described this on line 141-142. 

15. Also were weights not applied in the analysis to account for age and gender characteristics of the student population in the three universities? I know that the World Mental Health College Student Initiatives usually include weights in their studies? In the acknowledgements it mentions weights. If weights were used it should be mentioned in the methodology section.

Response: We have described how weighting was done (line 191-192). 

16. Explain what exactly this is - 12-month substance use disorder (other than alcohol). Is it drug use? If so, describe in details what it covers. This could be described in the methods section.

Response: We have described the substances that were assessed for other than alcohol (i.e. cannabis, cocaine, LSD, opioids, speed, ecstasy, prescription medication). Line 173-174

17. It is important to also report the number/prevalence of those who had any SUD in the text. Perhaps you could include co-morbid prevalence in Table 2.

Response: We have added this on Table 2 (line 221)

18. The results are somewhat confusing, and I am not sure why you run some of the analysis that you did. I think that it would be better to report the prevalence rates of mental health disorders for all participants and then compare these with the rates for those with SUD. This would be much more informative than what is presented in Table 4.

Response: We have removed Table 4 and left Table 5. 

19. That is the rationale for both the bivariate model and the regression model and why is it described as a multivariate logistic regression? A multivariate logistic regression is a model with more than one dependent variable. It the methodology it states that the dependent variable is just Any SUD. If this is the case, it would be an ordinary binary logistic regression. small sample size. Also were all the assumptions of a regression considered? This would be important given the small sample size. 

Response: The bivariate analysis done is the normal univariate analysis that is done using Fishers Exact test or Chi Squared test for comparing 2 or more groups with categorical data and Students t-test or Kruskal Wallis test for continuous data. Based on the univariate/bivariate analysis, a multivariate logistic regression model was run to identify associations between different mental disorders separately with SUD. For the multivariate logistic regression model, the dependent variable was SUD and the independent variables were all different mental disorders run separately with adjusting for age and gender. All assumptions were considered while running the models. 

We have further described this on line 198-202

20. It is really now clear how this model was run and further detail are required. Were all the IVs entered together? It is important to consider how this would impact on the findings.

Response: For the multivariate logistic regression model, the dependent variable was SUD and the independent variables were all different mental disorders run separately with adjusting for age and gender. We have clarified this on line 198-202

21. When reporting findings from a logistic regression it is normal to comment on the odds ratios rather than the p value. For example, students with depression were nearly three time more likely to have SUD (OR= 2.897). It is also normal to use 1.0 rather than reference and then place this below the ORs and Cis.

Response: We have revised and corrected this. (line 250-259)

Discussion

22. Again, it isn’t good practice to refer to ‘our’ study. Say this study or something similar. 

Response: We have reworded and revised it. (Line 271)

23. The first few lines are very repetitive of what was said in the introduction. It would be good to re-word.

Response: We have reworded and revised it. (Line 271-289)

24. 273 – People rather than persons

Response: We have corrected this. (line 311)

25. 275 – typo – remove of before mental illness.

Response: We have corrected this. (line 313)

26. It would be good to include further implications of the study in your conclusion.

Response: We have collated all implications into one section and expounded on them. (line 316-341)

Reviewer #2: 

27. It is very important for diagnosis by psychiatrists for both mental disorders and substance-related disorders. Thus, all data should be re-categorized and re-analyze to be problematic substance use and "depression", "anxiety", whatever the questionnaire should be interpreted.

Response: The tool used for the World Mental Health International College Student (WMH-ICS) Initiative is designed to generate accurate epidemiological data on mental, substance, and behavioral disorders among college students. Other studies conducted as part of the initiative have reported on epidemiology of disorders. The authors prefer to keep the diagnoses as disorders in this manuscript.

(https://www.hcp.med.harvard.edu/wmh/college_student_survey.php).

---

## [Decision Letter · Decision Letter 1]

4 Sep 2023

PONE-D-23-12172R112-month substance use disorders among first-year university students in KenyaPLOS ONE

Dear Dr. Jaguga,

Thank you for submitting your manuscript to PLOS ONE. After careful consideration, we feel that it has merit but does not fully meet PLOS ONE’s publication criteria as it currently stands. Therefore, we invite you to submit a revised version of the manuscript that addresses the points raised during the review process.

We look forward to receiving your revised manuscript.

Kind regards,

Md. Saiful Islam, BPH, MPH

Academic Editor

PLOS ONE

Journal Requirements:

Additional Editor Comments:

Thanks for addressing all comments. Please address the Reviewer 1 comments before accepting the manuscript.

Reviewers' comments:

Reviewer's Responses to Questions

**Comments to the Author**

1. If the authors have adequately addressed your comments raised in a previous round of review and you feel that this manuscript is now acceptable for publication, you may indicate that here to bypass the “Comments to the Author” section, enter your conflict of interest statement in the “Confidential to Editor” section, and submit your "Accept" recommendation.

Reviewer #1: All comments have been addressed

Reviewer #2: All comments have been addressed

2. Is the manuscript technically sound, and do the data support the conclusions?

Reviewer #1: Yes

Reviewer #2: Yes

3. Has the statistical analysis been performed appropriately and rigorously? 

Reviewer #1: Yes

Reviewer #2: Yes

4. Have the authors made all data underlying the findings in their manuscript fully available?

Reviewer #1: Yes

Reviewer #2: Yes

5. Is the manuscript presented in an intelligible fashion and written in standard English?

Reviewer #1: Yes

Reviewer #2: Yes

6. Review Comments to the Author

Reviewer #1: Did you check with Harvard about the sharing of data? I still don't think you can publically share but can note that.

Check English and grammer throughout manuscript - still reference to 'we' e.g. line 94, 136, 185, 262, 292, 303, 333, 335

Note the tables need to be formatted correctly - no vertical lines

In response to reviewer 2 it might be good to include something like the following:

Good concordance has been found between the WMH-CIDI and clinical assessments

Haro JM, Arbabzadeh-Bouchez S, Brugha TS, de Girolamo G, Guyer ME, Jin R, et al. Concordance of

the Composite International Diagnostic Interview Version 3.0 (CIDI 3.0) with standardized clinical

assessments in the WHO World Mental Health surveys. International journal of methods in psychiatric

research. 2006; 15(4):167–80. PMID: 17266013

Reviewer #2: (No Response)

7. PLOS authors have the option to publish the peer review history of their article (what does this mean?). If published, this will include your full peer review and any attached files.

Reviewer #1: No

Reviewer #2: **Yes: **Chonnakarn Jatchavala M.D.,MSc.

---

## [Author Response · Author response to Decision Letter 1]

15 Oct 2023

Response to reviewers’ and Editor’s comments

The authors thank the Editor and reviewers for the comments.

A point-by-point response to the comments has been provided below.

Reviewer #1: Did you check with Harvard about the sharing of data? I still don't think you can publically share but can note that.

Yes. We did check with Harvard, and they confirmed that since this paper, wasn’t based on cross-national data, but just Kenyan data, it was up to us to decide whether to make the data available or not. The authors have therefore decided to make the data available.

Comment: Check English and grammer throughout manuscript - still reference to 'we' e.g. line 94, 136, 185, 262, 292, 303, 333, 335

Response: The authors have restructured the sentences to remove ‘we’ and ensure clarity throughout the document.

Comment: Note the tables need to be formatted correctly - no vertical lines

Response: The vertical lines have been removed on all tables in the results section

Comment: In response to reviewer 2 it might be good to include something like the following:

Good concordance has been found between the WMH-CIDI and clinical assessments

Haro JM, Arbabzadeh-Bouchez S, Brugha TS, de Girolamo G, Guyer ME, Jin R, et al. Concordance of the Composite International Diagnostic Interview Version 3.0 (CIDI 3.0) with standardized clinical assessments in the WHO World Mental Health surveys. International journal of methods in psychiatric research. 2006; 15(4):167–80. PMID: 17266013

Response: Thank you for this comment. We have revised the response to reviewer 2 to read as follows:

The tool used for the World Mental Health International College Student (WMH-ICS) Initiative is designed to generate accurate epidemiological data on mental, substance, and behavioral disorders among college students. Other studies conducted as part of the initiative have reported on epidemiology of disorders. The authors prefer to keep the diagnoses as disorders in this manuscript. (https://www.hcp.med.harvard.edu/wmh/college_student_survey.php). 

In addition, good concordance has been found between the WMH-CIDI and clinical assessments.

(Haro JM, Arbabzadeh-Bouchez S, Brugha TS, de Girolamo G, Guyer ME, Jin R, et al. Concordance of the Composite International Diagnostic Interview Version 3.0 (CIDI 3.0) with standardized clinical assessments in the WHO World Mental Health surveys. International journal of methods in psychiatric research. 2006; 15(4):167–80. PMID: 17266013)

---

## [Decision Letter · Decision Letter 2]

26 Oct 2023

12-month substance use disorders among first-year university students in Kenya

PONE-D-23-12172R2

Dear Dr. Florence Jaguga,

We’re pleased to inform you that your manuscript has been judged scientifically suitable for publication and will be formally accepted for publication once it meets all outstanding technical requirements.

Kind regards,

Md. Saiful Islam, BPH, MPH

Academic Editor

PLOS ONE

Reviewers' comments:

Reviewer's Responses to Questions

**Comments to the Author**

1. If the authors have adequately addressed your comments raised in a previous round of review and you feel that this manuscript is now acceptable for publication, you may indicate that here to bypass the “Comments to the Author” section, enter your conflict of interest statement in the “Confidential to Editor” section, and submit your "Accept" recommendation.

Reviewer #1: All comments have been addressed

2. Is the manuscript technically sound, and do the data support the conclusions?

Reviewer #1: Yes

3. Has the statistical analysis been performed appropriately and rigorously? 

Reviewer #1: Yes

4. Have the authors made all data underlying the findings in their manuscript fully available?

Reviewer #1: Yes

5. Is the manuscript presented in an intelligible fashion and written in standard English?

Reviewer #1: Yes

6. Review Comments to the Author

Reviewer #1: I am happy that you addressed my previous comments. I think it would be good to include the reference to Haro et al in the actual manuscript but it's up to you.

7. PLOS authors have the option to publish the peer review history of their article (what does this mean?). If published, this will include your full peer review and any attached files.

Reviewer #1: No

---

## [Editor Report · Acceptance letter]

13 Nov 2023

PONE-D-23-12172R2 

12-month substance use disorders among first-year university students in Kenya 

Dear Dr. Jaguga:

I'm pleased to inform you that your manuscript has been deemed suitable for publication in PLOS ONE. Congratulations! Your manuscript is now with our production department. 

Kind regards, 

on behalf of

Dr. Md. Saiful Islam 

Academic Editor

PLOS ONE